# Osimertinib in the Treatment of Epidermal Growth Factor Receptor-Mutant Early and Locally Advanced Stages of Non-Small-Cell Lung Cancer: Current Evidence and Future Perspectives

**DOI:** 10.3390/cancers17040668

**Published:** 2025-02-16

**Authors:** Antonello Veccia, Mariachiara Dipasquale, Martina Lorenzi, Sara Monteverdi, Stefania Kinspergher, Elena Zambotti, Orazio Caffo

**Affiliations:** Medical Oncology, Santa Chiara Hospital, Largo Medaglie d’Oro 1, 38122 Trento, Italy; mariachiara.dipasquale@apss.tn.it (M.D.); martina.lorenzi@apss.tn.it (M.L.); sara.monteverdi@apss.tn.it (S.M.); stefania.kinspergher@apss.tn.it (S.K.); elena.zambotti@apss.tn.it (E.Z.); orazio.caffo@apss.tn.it (O.C.)

**Keywords:** osimertinib, early stage, adjuvant, maintenance, neoadjuvant, resistance

## Abstract

Osimertinib is a third-generation inhibitor of epidermal growth factor receptor (EGFR), used for treating metastatic, locally advanced, and early-stage non-small-cell lung cancer (NSCLC) with EGFR mutations. In this review, we focus on the clinical studies that led to the introduction of osimertinib in locally advanced and early-stage NSCLC. Specifically, we describe the studies that support osimertinib use as adjuvant treatment in radically resected NSCLC patients, and as maintenance therapy in patients previously treated with concomitant chemoradiotherapy. We also report the ongoing clinical trials on osimertinib in neoadjuvant setting, alone or in combination with other drugs. Moreover, we address a number of issues about the use of osimertinib in locally advanced and early stages, such as the efficacy of the drug on uncommon mutations, the long-term impact on survival, and the management of resistance mechanisms. Finally, we describe the ongoing clinical trials focusing on research that aims to identify biomarkers that are more able to select patients who will benefit most from osimertinib treatment.

## 1. Introduction

The treatment of non-small-cell lung cancer (NSCLC) patients with epidermal growth factor receptor (EGFR) mutations radically changed after the introduction of EGFR tyrosine kinase inhibitors (TKIs) to clinical practice, both in a metastatic setting [1,2] and in earlier stages of disease [3,4]. Data from the literature report that EGFR mutations are detected in about 13% and 50% of Caucasian and Asian NSCLC patients, respectively [5]. They are more frequent in non-squamous histology, in women, and in never or light smokers [6]. EGFR mutations are spatially located within the ATP binding site of the kinase and show different sensitivity to EGFR inhibitors [7]. Most EGFR mutations (85–90%) are in-frame deletions of exon 19 and mutations of exon 21 L858R: they are common, known as activating mutations, and show high sensitivity to EGFR TKIs [7,8]. The remaining mutations (10–15%) include exon 18 point mutations, exon 21 L861X, exon 20 insertions, and exon 20 S768I: they are associated with shorter survival and show different sensitivity to TKIs [9,10]. Specifically, tumors with insertions in exon 20 have a prognosis similar to wild-type tumors and respond better to chemotherapy [11,12]. Moreover, a secondary point mutation that substitutes methionine for threonine at amino acid position 790 (T790M) has emerged as the main acquired resistance mechanism to EGFR TKIs [13].

Over the years, several TKIs have been developed to target EGFR mutations in metastatic EGFR-mutant NSCLC. First-generation TKIs gefitinib and erlotinib showed significantly longer PFS over chemotherapy in NSCLC patients with EGFR common activating mutations [14,15]. Then, second-generation TKIs afatinib and dacomitinib reported better survival compared to first-generation TKIs [16,17,18]. Finally, the third-generation TKI osimertinib was initially registered for patients who developed the EGFR T790M mutation during or after EGFR TKI treatment because of its advantage in survival over chemotherapy [19], then as first-line treatment in untreated patients, reporting a statistically significant benefit in survival compared to first-generation TKIs [20,21] (Table 1).

Based on the results reported in metastatic disease, osimertinib has been investigated in earlier stages of disease, also confirming its significant impact on survival in these stages and becoming the new standard of care in international guidelines [3,22].

We reviewed the scientific evidence and clinical impact of osimertinib as adjuvant treatment in completely resected EGFR-mutant NSCLC, as maintenance treatment after chemoradiotherapy in locally advanced disease, and as a preoperative treatment in the neoadjuvant setting. Moreover, we discuss the open questions related to the management of osimertinib, including the resistance mechanisms that can develop during treatment (Figure 1). These mechanisms can be intrinsic or acquired, with the latter classified into EGFR-dependent (or on-Target) and EGFR-independent (or off-Target). An accurate detection of these mechanisms through tissue rebiopsy or liquid biopsy allows us to define the best option and customize it for each patient.

## 2. Osimertinib in the Adjuvant Setting

About 30% of NSCLC cases are resectable at diagnosis. For many years, after surgery, chemotherapy was the only postoperative treatment recommended by the guidelines for pathological stages IIB and III (according to TNM 8th edition), while it was only to be considered for stage IIA (T ≥ 4 cm) [3,4,23]. However, the survival benefit was limited and estimated to be 4–5% at 5 years in terms of disease-free survival (DFS) and overall survival (OS). In recent years, adjuvant treatment has been completely revolutionized by the introduction of immunotherapy and osimertinib in EGFR wild-type and -mutant patients, respectively. Osimertinib has been registered as adjuvant treatment in EGFR-mutant patients with postoperative stages II to IIIA [3,4,23].

The ADAURA study is a phase 3, double-blind, placebo-controlled, international trial that randomized completely resected NSCLC patients with common EGFR mutations to receive, with a 1:1 ratio, either oral osimertinib (at a dose of 80 mg once daily) or placebo for 3 years, with or without previous adjuvant chemotherapy. The patients were stratified according to disease stage (IB, II, or IIIA), EGFR mutational status (exon 19 deletion or L858R), and race (Asian or non-Asian). The primary and secondary endpoints are reported in Table 2. From November 2015 to February 2019, 682 patients were enrolled: 339 received osimertinib and 343 received placebo. Baseline patients’ characteristics were well balanced between the two groups. In the osimertinib arm, most of the patients were female (68%), Asian (64%), and never smokers (68%), with a median age of 64 years and diagnosis of adenocarcinoma (96%); the postoperative stage was IB, II, and IIIA in 32%, 34%, and 35% of patients, respectively; 60% of patients had positive lymph nodes and received adjuvant chemotherapy. At data cut-off in January 2020, after a median follow-up of 22.1 months, the study reached its primary endpoint, with a significantly longer DFS among patients with stage II to IIIA in the osimertinib group compared to those who received placebo (HR 0.17; 96.06% CI, 0.11–0.26; *p* < 0.001). The median DFS was not reached (95% CI, 38.8–NR) in the osimertinib group vs. 19.6 months (95% CI, 16.6–24.5) in the placebo group. A benefit in terms of DFS was also observed in the overall population (HR 0.20; 99.12% CI, 0.14–0.30; *p* < 0.001) and in all subgroups of patients, regardless of stage and previous use of chemotherapy [24].

These results were confirmed at the following data cut-off, in April 2022. After a median follow-up of 44.2 months, DFS HR was 0.27 (95% CI, 0.21–0.34) and 0.23 (95% CI, 0.18–0.30) in the overall and stage II and III populations, respectively. The rate of patients experiencing local/regional and distant recurrence was lower in patients treated with osimertinib than in those who received placebo. Stage II-IIIA patients treated with osimertinib had a 70% chance of being disease-free at 4 years compared to 29% of those treated with placebo. Moreover, in the same population, osimertinib treatment was associated with a lower rate of central nervous system relapse (CNS DFS HR 0.24; 95% CI, 0.14–0.42) [25].

In 2023, mature OS data were published: in the overall population, the 5-year OS was 88% in the osimertinib group and 78% in the placebo group (HR, 0.49; 95.03% CI, 0.34–0.70; *p* < 0.001). In the population with stage II-IIIA disease, the 5-year OS was 85% (95% CI, 79–89) in the osimertinib group and 73% (95% CI, 66–78) in the placebo group (HR 0.49; 95.03% CI, 0.33–0.73; *p* < 0.001) [26].

Regarding toxicity, osimertinib showed a good safety profile (Table 2). The most common any-grade toxicities associated with osimertinib were diarrhea (46%), paronychia (25%), dry skin (23%), and pruritus (19%). Interstitial lung disease was reported in 3% and 0% of the patients in the osimertinib and placebo groups, respectively.

Table 2 summarizes dose interruptions, dose reductions, and discontinuation due to adverse events [24,27].

Ongoing clinical trials with osimertinib in the adjuvant setting for resected EGFR-mutant NSCLC are shown in Table 3.

### Open Questions

Despite the results reported by the ADAURA study, many issues remain unresolved.

Firstly, we will discuss the role of adjuvant chemotherapy. In the ADAURA study, chemotherapy was administered mainly in patients aged ≤ 70 years and with stages II or IIIA; in addition, 60% of patients received platinum plus vinorelbine or pemetrexed. Moreover, patients were not stratified according to previous chemotherapy use. Thus, if we know that the benefit of osimertinib is independent of prior administration of chemotherapy, we have no basis for deciding which patients should receive chemotherapy and osimertinib and which should receive only osimertinib. Osimertinib alone, without previous chemotherapy, might be proposed to patients who are frail for age or based on comorbidities and ECOG performance status. On the other hand, the use of previous chemotherapy should be considered in patients at higher risk of relapse, such as those with positive lymph nodes.

Secondly, the optimal duration of the treatment should be considered. In the ADAURA study, only 66% of patients completed the planned 3 years of treatment with osimertinib. It could be assumed that the longer the duration of treatment, the better the disease control with reduced risk of recurrence and improved DFS and OS. Therefore, the TARGET study has been designed and is currently ongoing. It is a phase 2, multinational, open-label, single-arm trial on resected stage II to IIIB NSCLC patients with common (exon 19 deletion or L858R) or uncommon (G719X, L861Q, and/or S768I) EGFR-TKI sensitizing mutations. They receive osimertinib 80 mg once daily for 5 years or until disease recurrence, discontinuation, or death. Prior adjuvant chemotherapy is allowed. The primary endpoint is 5-year DFS in the cohort with common mutations. Secondary endpoints include 5-year DFS in the cohort with uncommon mutations; OS at 3, 4, and 5 years in the overall population; safety and tolerability; type of recurrence; and CNS metastases (both cohorts). The results of this study are expected in 2029 [28] (Table 3).

Thirdly, the efficacy of osimertinib on uncommon mutations is noteworthy. In ADAURA, only patients with common mutations (del 19 and L858R) received adjuvant osimertinib; therefore, the use of osimertinib in uncommon mutations is not recommended. In a metastatic first-line setting, although patients with uncommon mutations were excluded from the FLAURA study [20], evidence from real-world practice [29,30] and prospective phase 2 [31] studies confirmed the efficacy of osimertinib in this subgroup of patients. This suggests that osimertinib could also be active in patients with uncommon mutations in the adjuvant setting, but prospective data are awaited.

The prognostic role of co-mutations represents a further factor influencing the efficacy of osimertinib in the adjuvant setting. Data from ADAURA are lacking but literature data report that protein 53 (p53) is the most frequently co-mutated gene in early-stage EGFR-mutant NSCLC [32,33], and it is also associated with a poor prognosis in this setting [34].

However, p53 does not have a predictive role but only a prognostic one, and therefore, p53 positivity alone cannot be considered sufficient to exclude a patient from treatment with osimertinib.

Another issue is the role of adjuvant osimertinib in earlier stages. The advantage of osimertinib was reported in all stages in the ADAURA study, starting from tumors ≥ 3 cm, although it was more limited. ADAURA2 is a phase 3, randomized, double-blind, placebo-controlled study evaluating adjuvant osimertinib vs. placebo in patients with stage IA2-IA3 EGFR-mutant NSCLC after surgical resection, stratified for pathologic risk of disease recurrence (high vs. low), EGFR mutation type (exon 19 deletion vs. L858R), and race (Chinese Asian vs. non-Chinese Asian vs. non-Asian). Patients will receive osimertinib 80 mg/day or placebo until disease recurrence, treatment discontinuation, or a maximum treatment duration of 3 years. The primary endpoint is DFS in the high-risk group. Secondary endpoints include DFS in the overall population, overall survival, CNS DFS, and safety. The study is currently recruiting and interim results of the primary endpoint are expected in August 2027 [35]. In the meantime, stage I patients should not be offered adjuvant osimertinib.

A further topic is how to address the selection for adjuvant osimertinib. At 48 months, the percentage of patients alive without disease was 73% (95% CI, 67 to 78) for osimertinib and 38% (95% CI, 32 to 43) for placebo. This means that about a third of patients can be considered cured even without receiving osimertinib. Therefore, it is essential to identify the patients at high risk for relapse who may benefit from adjuvant treatment. The most promising biomarker of relapse is minimal residual disease (MRD) based on the detection of circulating tumor DNA (ctDNA) after surgery [36,37,38,39,40,41].

Data on ctDNA from the ADAURA study were presented at the ASCO annual meeting in 2024. Among 682 randomized patients, 32% (220) of patients across both arms had samples evaluable for MRD. At baseline, 5/112 (4%) in the osimertinib arm and 13/108 (12%) in the placebo arm were MRD-positive: 4/5 (80%) of patients treated with osimertinib vs. none of 13 patients receiving placebo became MRD-negative. MRD detection had clinical sensitivity and specificity of 65% and 95%, respectively, and preceded a DFS event by a median of 4.7 (95% CI, 2.2–5.6) months across both arms. At 3 years, 86% of patients treated with osimertinib vs. 36% of those with placebo were MRD-negative and without disease (HR: 0.23; 95% CI, 0.15–0.36). These data support the potential use of MRD detection to identify patients likely to benefit from longer adjuvant osimertinib [42].

CTONG 2201 is an ongoing prospective, multicenter, and single-arm study aimed to demonstrate that NSCLC patients in pathological stage IB to IIIA do not need adjuvant treatment when longitudinal MRD is undetectable after surgery [43].

ECTOP-1022 (NCT06323148) is a phase 3 study that will evaluate the use of adjuvant osimertinib vs. observation in ctDNA-positive patients with stage II–IIIA NSCLC and EGFR mutation.

A final consideration concerns treatments after progression during or after completion of adjuvant osimertinib. In the ADAURA study [26], 76 patients (22%) in the osimertinib arm received a subsequent anticancer treatment compared to 184 (54%) of those in the placebo arm. In the osimertinib arm, 58 (76%) patients received an EGFR TKI: osimertinib was the preferred drug for 31 (41%) of the cases, while only 20 patients (26%) received chemotherapy. In the literature, no data deriving from phase 3 clinical trials are available to guide the choice of treatment at disease recurrence. We know that several resistance mechanisms may develop through EGFR TKIs [44,45,46], but these data mainly derive from patients treated in first or subsequent lines of treatment. Two case reports described a successful osimertinib rechallenge in patients who relapsed after adjuvant osimertinib [47], in line with results reported by a phase 2 trial of erlotinib rechallenge in patients who relapsed after adjuvant erlotinib [48]. However, this approach is applicable only when an adequate time interval has elapsed since the completion of adjuvant osimertinib. When the relapse occurs during the treatment, chemotherapy should be proposed, or alternatively, the subsequent treatment should be guided by the identification of the specific resistance mechanism responsible for disease progression. We know, however, that the latter approach is made difficult by the limited availability of NGS at all cancer centers and the difficulty in accessing drugs for which the patient might be a candidate after biomolecular recharacterization. The NCT06530719 study is an active study that is not yet recruiting aiming to verify the efficacy and safety of osimertinib rechallenge in patients with EGFR-mutant NSCLC, who progressed after adjuvant targeted therapy following radical surgery.

ROSIE (NCT06053099) is an ongoing prospective cohort study designed to evaluate molecular prognostic factors and resistance mechanisms to osimertinib as adjuvant treatment of completely resected IB-IIIA NCSLC with common EGFR mutations. The evaluation is performed by centralized next-generation sequencing (NGS) on both ctDNA and formalin-fixed paraffin-embedded (FFPE) tissue. RAISE (NCT06477055) is another prospective and multicentric phase 2 study that will evaluate the recurrence gene profiles of adjuvant osimertinib in resected EGFR-mutant NSCLC.

## 3. Osimertinib in Locally Advanced NSCLC

Patients in stage III represent about 30% of NSCLC cases at diagnosis. Most of these patients have unresectable disease and are candidates for concomitant chemoradiotherapy treatment followed by durvalumab maintenance for one year [3,4,22,23], according to the results of the PACIFIC study [49,50]. A post hoc exploratory analysis of this study reported efficacy and safety data of 35 EGFR-mutant patients, 24 receiving durvalumab and 11 placebo: no differences were reported in terms of PFS and OS [51]. However, these data must be interpreted with caution due to the small number of patients included in the analysis; moreover, most patients were men (54%) and smokers (54%). Although real-word data also confirmed a potential role of durvalumab maintenance in EGFR-mutant patients [52], over the years, the strategy based on maintenance with TKI in EGFR-mutant patients has been considered the most promising and worthy of study.

The LAURA study is a phase 3, double-blind, placebo-controlled trial that randomized (with 2:1 ratio) patients with unresectable stage III NSCLC, with EGFR exon 19 deletion or exon 21 L858R point mutation, and without progression during or after chemoradiotherapy, to receive osimertinib or placebo until disease progression. The primary and secondary endpoints are reported in Table 4. From August 2018 to July 2022, 216 patients were enrolled: 143 received osimertinib and 73 received placebo. Baseline patients’ characteristics were well balanced between the two groups. In the osimertinib arm, most patients were female (63%), Asian (81%), and never smokers (71%), with a median age of 62 years and diagnosis of adenocarcinoma (97%); the stage was IIIA, IIIB, and IIIC in 36%, 47%, and 17% of patients, respectively; 52% of patients had exon 19 mutation and 48% had L858R mutation; radiotherapy was concurrent to chemo in 92% of cases. At the data cut-off in January 2020, after a median follow-up of 22 months, PFS was significantly longer in patients treated with osimertinib compared to those receiving placebo (39.1 vs. 5.6 months; HR for PFS: 0.16; 95% CI, 0.10–0.24; *p* < 0.001). At 1 and 2 years, 74% and 65% of patients of the experimental arm were alive versus 22% and 13% of the placebo arm, respectively. A benefit in terms of PFS was observed in all subgroups of patients. Lower incidences of local progression and distant metastases were observed in the osimertinib group (21% and 16%, respectively) than in the placebo group (48% and 37%, respectively). Data on response rate, median duration of response, and adverse events are shown in Table 4. The most common observed side effects were radiation pneumonitis (48% with osimertinib vs. 38% with placebo), diarrhea (36% vs. 14%), and rash (24% vs. 14%) [53].

An analysis from the LAURA study concerning the impact of osimertinib on CNS disease control has been recently published. Osimertinib was associated with fewer CNS PFS events compared with placebo: 29/143 (20%) vs. 30/73 (41%), respectively. In both arms, most CNS PFS events were due to CNS progression (13% vs. 36% in osimertinib and placebo, respectively). Patients treated with osimertinib showed a median CNS PFS significantly longer than those treated with placebo—not reached vs. 14.9 months (95% CI 7.4–NC)—corresponding to 83% reduction in the risk of CNS progression or death with osimertinib (HR 0.17; 95% CI 0.09–0.32, *p* < 0.001). The cumulative incidence of CNS progression was lower with osimertinib vs. placebo: 9% vs. 36% at 1 year, and 12% vs. 37% at 2 years. Also, the percentage of patients with a death or distant metastasis event was lower in the osimertinib arm (23% vs. 42%). The median time to development of distant metastases (TTDM) was not reached (95% CI 39.3 month–NC) with osimertinib versus 13.0 months (95% CI 9.0 month–NC) with placebo; HR 0.21 (95% CI 0.11–0.38; *p* < 0.001). This translated into a 79% reduction in the risk of distant metastases or death with osimertinib vs. placebo. The cumulative incidence of distant metastases was lower with osimertinib vs. placebo: 11% vs. 37% at 1 year, and 14% vs. 39% at 2 years [54].

A recent retrospective study comparing osimertinib and durvalumab as maintenance after chemoradiotherapy in patients with stage III unresectable NSCLC and sensitizing *EGFR* mutation suggested that consolidation osimertinib was associated with a significantly longer PFS [55].

Ongoing clinical trials with osimertinib in locally advanced EGFR-mutant NSCLC are shown in Table 5.

### Open Questions

Data from the LAURA study have completely revolutionized the treatment of patients with EGFR-mutated locally advanced NSCLC, becoming the new standard treatment in these patients. However, some issues need to be further explored.

Similarly to the ADAURA study, LAURA also only included patients with common mutations. Therefore, the benefit of osimertinib in uncommon mutations will have to be better investigated. Real-world data will likely support the efficacy of osimertinib and its use in uncommon mutations, but data from prospective studies are necessary to confirm this approach.

A second issue is the long-term efficacy of osimertinib. To date, data on OS from the LAURA study are not available because they are immature due to the short follow-up. At interim analysis of OS, with a data maturity of 20%, no significant difference between osimertinib and placebo has been found in terms of OS [53].

According to the study design, osimertinib was administered until disease progression, but it is not known whether all patients need to take the drug until progression. In this sense, data on ctDNA would be useful to quantify the residual disease and identify which patients truly benefit from administering osimertinib until progression. This aspect is very important because continuous treatment is associated with the development of resistance.

In addition, another question is how to manage the disease progression. In the LAURA trial, osimertinib was offered to patients in both treatment arms after disease progression. In the placebo group, 81% of patients who developed disease progression received subsequent osimertinib treatment. In the osimertinib group, as osimertinib could be continued if a clinical benefit was maintained, 28% of patients continued osimertinib beyond progression. Clearly, without robust evidence, in clinical practice, the choice of maintaining the drug beyond progression rather than switching to a second line of chemotherapy requires a careful evaluation of the risk–benefit ratio.

## 4. Osimertinib in the Neoadjuvant Setting

In recent years, several randomized clinical trials have investigated the role of immune checkpoint inhibitors (ICIs) in combination with chemotherapy as neoadjuvant treatment in resectable NSCLC [56,57,58], reporting a statistically significant prolongation of event-free survival (EFS) and increase in pathologic complete response (pCR) rates, which has been recognized as a surrogate of OS [59,60]. However, all these studies except KEYNOTE-671 [61] excluded patients with EGFR mutations, who have been enrolled in clinical trials specifically designed for EGFR-mutant disease [62].

The NEOS study was a single-arm phase 2 trial that studied osimertinib as neoadjuvant treatment in 40 patients with stage IIA-IIIB lung adenocarcinoma and classic EGFR mutations. The ORR was 71.1% in 38 patients who completed the 6-week osimertinib treatment; 32 patients underwent surgery, with negative margins in 30 patients (93.8%). Although 75% of patients developed adverse events, the toxicity was manageable: the most common were rash (50%), diarrhea (30%), and oral ulceration (30%) [63].

In another phase 2 trial, patients with stage I-IIIA EGFR-mutant (del 19 or L858R) NSCLC received osimertinib 80 mg/day for up to two 28-day cycles before undergoing surgery. A total of 27 patients were treated for a median of 56 days before surgery, which was performed in 89% of cases. The major pathological response (MPR), defined as the portion of surviving tumor cells in different parts of the tumor found during surgery after neoadjuvant therapy, was 14.8%; no pCRs were observed, but 96% of patients achieved partial response or stable disease. The median DFS after surgical resection was 32 months (95% CI; 26 not reached) with a median follow-up of 11 months. However, the primary endpoint of 50% MPR was not reached [64].

NeoADAURA is a phase 3 study evaluating the efficacy and safety of neoadjuvant osimertinib alone or in combination with chemotherapy versus chemotherapy alone in patients with stage II-IIIB NSCLC and common EGFR mutations, followed by surgery and adjuvant treatment. The primary endpoint of the study is to evaluate the MPR at resection; secondary endpoints include EFS, pCR, lymph nodal downstaging at the time of resection, DFS, OS, health-related quality of life (HRQoL), MPR in patients with/without EGFRm detectable at screening in ctDNA, and concordance between baseline tumor DNA and ctDNA. The study is estimated to enroll 351 patients in order to detect a statistically significant difference in MPR of 20%. The study is ongoing [65].

Neoadjuvant osimertinib was also investigated for downstaging the disease before radiation therapy.

A nonrandomized, single-arm, phase 2 prospective trial enrolled patients with inoperable, stage III EGFR-mutant NSCLC who received osimertinib 80 mg daily for 12 weeks. Responder patients underwent definitive radiation therapy (RT) and/or surgery; non-responders received standard CRT; then, patients were followed for 2 years without adjuvant therapy. The primary endpoint was the objective response rate (ORR); secondary endpoints were safety and parameters regarding the variation in tumor volume, before versus after osimertinib. The study included 24 patients with stage IIIA-C, most of them with adenocarcinoma. The ORR to osimertinib was 95.2%; after induction osimertinib, 13 and 3 of 20 patients underwent radiotherapy and surgery, respectively. Moreover, osimertinib induction significantly reduced the radiation field by nearly 50% with a linear association with tumor size [66].

Finally, osimertinib is also currently being studied as consolidation treatment after stereotactic body radiation therapy (SBRT). PACIFIC-4 is an international, phase 3 study including a cohort of 60 EGFR-mutant NSCLC patients with localized disease who are candidates for SBRT followed by osimertinib 80 mg per os daily for up to 36 months. Inclusion criteria are common EGFR mutation (ex19del or L858R), unresected stage I-II (T1-T3, N0, M0), and ECOG PS 0–2. The primary endpoint is 4-year PFS; secondary endpoints are OS and safety. The trial is recruiting [67] (Table 5).

Other ongoing clinical trials with osimertinib as neoadjuvant treatment for EGFR-mutant NSCLC are shown in Table 5.

### Open Questions

Neoadjuvant osimertinib is associated with several benefits in EGFR-mutant NSCLC: it can downstage the tumor and make it operable; moreover, an early initiation of osimertinib may reduce the risk of distant metastases. However, some disadvantages need to be highlighted: first of all, the risk of delaying the surgical intervention or developing surgical complications, such as adhesions or fibrosis; in addition, a percentage of patients do not respond to osimertinib treatment and never undergo surgery. Secondly, from a methodological point of view, the studies on neoadjuvant osimertinib have some weaknesses: the pathological response has not yet been validated as a surrogate of OS in lung cancer, and mature data on long-term survival are lacking. Furthermore, data deriving from these studies are sometimes discordant: despite high ORR, the endpoint of MPR was not always reached.

Another unsolved issue is the comparison between neoadjuvant and adjuvant treatment: in fact, in the absence of randomized comparison studies, it is not possible to define whether neoadjuvant osimertinib is superior to adjuvant osimertinib.

ctDNA will likely have an essential role during neoadjuvant therapy for identifying patients with postoperative minimal residual disease who receive limited benefits from osimertinib alone and need to be escalated to systemic therapy [68].

Neo-Bio-ADAURA (NCT06206850) is a single-arm phase II study that will enroll 20 patients from the NeoADAURA study, whose tumor specimens including tissue biopsies will be used to identify molecular mechanisms of resistance to treatment.

## 5. Mechanism of Resistance to Osimertinib

The development of resistance represents one of the main problems when treating EGFR-mutant NSCLC. This topic has already been partially addressed in the sections on the use of osimertinib in the adjuvant and locally advanced setting. In particular, suggestions have been given on the clinical management of the patients, based purely on clinical assessments, such as the onset of progression during or after the end of osimertinib therapy. Instead, the most appropriate approach should be one based on the biomolecular re-characterization of the disease, so that tailored treatment can be defined on the basis of the molecular mechanism that caused the progression.

We do not yet fully understand the mechanisms of resistance that develop when osimertinib is used in the earlier stages of disease and, to date, we use data derived from metastatic disease. However, we know that resistance mechanisms differ in terms of both incidence and type even if the drug is used as first- or second-line treatments in pre-treated patients [44,45,46,69]. Similarly, these mechanisms could be different when osimertinib is used in earlier stages of NSCLC. Resistance to osimertinib can be intrinsic or acquired; the latter can be classified into EGFR-dependent (or on-Target) and EGFR-independent (or off-Target). Intrinsic resistance mechanisms develop within 6 months of starting osimertinib and concern about 20–30% of patients. They include exon 20 insertions, uncommon EGFR mutations other than exon 20 insertions, and co-occurring genomic alterations such as TP53, MET, RB1, ERBB2, KRAS, and PIK3CA [69]. In recent years, the tyrosine kinase receptor AXL was also found to be associated with tumor progression, epithelial-to-mesenchymal transition (EMT), and primary resistance to osimertinib: it accelerates the emergence of T790M and its levels positively correlate with the mutational loads of patients’ tumors [70,71,72]. AXL additionally drives the resistance to osimertinib of drug-tolerant cells by protecting them against treatment-induced DNA damage through the activation of low-fidelity DNA polymerases, MYC activation, and a pyridine/pyrimidine metabolism imbalance [73].

There is great interest in the use of ctDNA to assess AXL and high-fidelity DNA polymerase levels in osimertinib-resistant tumors and the possible therapeutic implications based on the use of drug combinations capable of targeting EGFR and AXL and delaying osimertinib resistance [74].

The two more frequent EGFR-dependent resistance mechanisms are T790M loss and C797S mutation. The concomitant development of C797S in trans and T790M restores sensitivity to the combination of first-generation EGFR TKI and osimertinib. EGFR-independent resistance mechanisms include MET alterations (such as amplification or exon 14 skipping), HER2 amplification, KRAS mutations, BRAF alterations (mutations or rearrangements), PTEN loss, and PIC3AK alterations [44,45,69].

Recent data suggest that the apolipoprotein B mRNA-editing enzyme catalytic polypeptide-like (APOBEC) mutational signature, deriving from a routine multi-panel test, can provide genomic information on resistance mechanisms to osimertinib and guide interventional strategies to overcome it [75,76].

Preclinical data and clinical reports support the use of osimertinib in combination with another inhibitor when a targetable oncogenic alteration (mutation, amplification, or fusion) is detected: for example, a combination of osimertinib and MET or BRAF inhibitors is a reasonable strategy to overcome osimertinib resistance attributed to MET amplification or BRAF mutations, respectively [44,45,46,69].

However, these therapeutic approaches have not been evaluated in randomized clinical trials; in addition, about 30–50% of resistance mechanisms to osimertinib remain unknown. Therefore, chemotherapy still represents the standard treatment when progression to osimertinib occurs. Moreover, 3–15% of patients may develop histological transformation, mainly to small-cell carcinoma [44,45,46,69]. This highlights the relevance of obtaining a contemporary sample by a tissue rebiopsy whenever possible; otherwise, a liquid biopsy can be performed.

## 6. Conclusions

The treatment of EGFR-mutant NSCLC has completely changed after the introduction of osimertinib in clinical practice, both in metastatic disease and in locally advanced/early stages. The first goal we must achieve is for all patients to be tested for EGFR: indeed, given the availability of osimertinib in different disease settings, it is unacceptable for the choice of treatment to be made without knowing the mutational status of EGFR. Based on the data of the ADAURA study, we must consider osimertinib as a new standard treatment in the adjuvant setting for patients with radically resected NSCLC and postoperative stages II to IIIA. In fact, the OS benefit of osimertinib is not only statistically significant but, above all, clinically relevant. Patients who are frail due to age and those with comorbidities and ECOG PS should receive osimertinib alone, while those at higher risk of relapse, such as those with positive lymph nodes, might benefit from prior adjuvant platinum-based chemotherapy. The use of osimertinib should not be limited to patients with common EGFR mutations but should also be extended to patients with rare mutations. To date, the p53 co-mutation only has a prognostic role; therefore, its detection cannot be considered sufficient to exclude a patient from treatment with osimertinib.

According to the LAURA study, osimertinib must also be considered the new standard for locally advanced EGFR NSCLC as maintenance treatment in patients who have not progressed after concurrent chemoradiotherapy. The benefit of osimertinib is incredibly high, not only in terms of survival but also in delaying the development of brain metastases. This represents a major success considering the high risk of brain metastases in EGFR-mutant patients. Also in this setting, we recognize a series of unsolved questions, such as the efficacy of the drug on uncommon mutations, the long-term impact on survival, and the management of resistance. On this last point, we suggest that, in the absence of data from randomized studies, it is reasonable to continue osimertinib when oligoprogression occurs rather than switching to a second line of chemotherapy, making a careful evaluation of the risk–benefit ratio. Finally, osimertinib has been investigated in the neoadjuvant setting, although data from clinical studies are controversial because only a few studies have achieved the primary endpoint of MPR. Therefore, to date, this approach must still be considered experimental. Moreover, also in this setting, some issues need to be clarified, such as the risk of delaying or not undergoing surgery for a limited number of patients and the role of pathologic complete response and event-free survival as a surrogate of overall survival. ctDNA represents the most promising biomarker to monitor the response to osimertinib when it is used as maintenance therapy or in the neoadjuvant setting and to define residual disease in resected patients. However, to date, its use in clinical practice requires appropriate validation before becoming a standard approach.

## 7. Future Directions

Several clinical studies are investigating osimertinib to resolve open questions about its use in early-stage EGFR-mutant NSCLC. The TARGET study will try to define the optimal duration of adjuvant treatment, comparing 5 years to 3 years of therapy in stage II-IIIB following surgical intervention. ADAURA2 is an ongoing phase 3 trial testing the efficacy of 3-year adjuvant osimertinib vs. placebo in stage I, similarly to the OSTAR study, which is a single-arm study evaluating DFS in stage I patients with high-risk factors (Table 3). Translational studies are needed to better define the potential use of ctDNA, both in tissue and blood, in detecting minimal residual disease and to identify patients at higher risk of recurrence that may benefit most from adjuvant osimertinib. In the unresectable stage III setting, it will be important to select patients who need to receive osimertinib until disease progression vs. those who can stop the treatment after an adequate duration. Valuable aid can come from ctDNA in this context. Moreover, the NEOLA study is evaluating the impact of osimertinib given before and after concurrent chemoradiotherapy on response and survival. In the neoadjuvant setting, several studies are evaluating the impact of osimertinib alone or in combination with other agents, such as aspirin or chemotherapy, on pathologic response after surgery, in addition to survival outcomes (Table 5). Many translational studies are also active in this setting. Patients with negative ctDNA after surgical intervention could avoid receiving adjuvant osimertinib, saving unnecessary toxicities. Moreover, the NeoADAURA study is collecting tissue biopsies in order to identify molecular mechanisms of resistance to treatment. This represents the most important issue in EGFR-mutant patients receiving osimertinib: this translational research, identifying the resistance mechanisms, may guide clinicians to choose the best treatment for their patients.

## Figures and Tables

**Figure 1 cancers-17-00668-f001:**
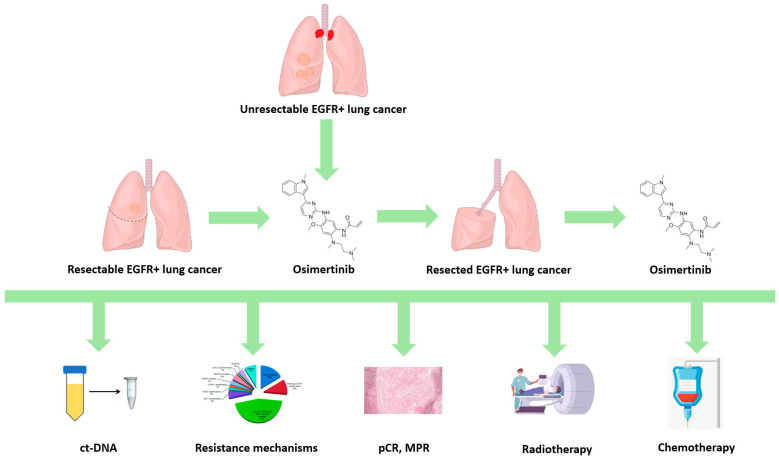
Open questions on osimertinib in early-stage EGFR-mutant NSCLC. EGFR = epidermal growth factor receptor; ctDNA = circulating tumor DNA; pCR = pathologic complete response; MPR = major pathologic response.

**Table 1 cancers-17-00668-t001:** Randomized phase 3 trials of first-, second-, and third-generation TKIs in metastatic NSCLC.

Trial	Phase	N. of Patients	EGFR TKI	Comparator Arm	EGFR Status	ORR (%)	mPFS (Months)	mOS (Months)
IPASS NCT00322452	3	1217	Gefitinib	Carboplatin + paclitaxel	all	71.2 vs. 47.3	9.5 vs. 6.3	21.6 vs. 21.9
EURTAC NCT00446225	3	173	Erlotinib	Cisplatin + docetaxel	mEGFR	64 vs. 18	9.7 vs. 5.2	19.3 vs. 19.5
LUX Lung 7 NCT01466660	2b	319	Afatinib	Gefitinib	mEGFR	72.5 vs. 56	11 vs. 10.9	27.9 vs. 24.5
ARCHER NCT01774721	3	452	Dacomitinib	Gefitinib	mEGFR	75 vs. 70	14.7 vs. 9.2	34.1 vs. 27
AURA 3 NCT021511981	3	419	Osimertinib	Carboplatin/cisplatin + pemetrexed	T790M mutated	71 vs. 31	10.1 vs. 4.4	26.8 vs. 22.5
FLAURA NCT02296125	3	556	Osimertinib	Gefitinib or erlotinib	mEGFR	80 vs. 76	18.9 vs. 10.2	38.6 vs. 31.8

All = mutant and wild-type EGFR; mEGFR = mutant EGFR; ORR = objective response rate; mPFS = median progression-free survival; mOS = median overall survival.

**Table 2 cancers-17-00668-t002:** Summary of ADAURA study in adjuvant NSCLC.

Phase	Stage	Treatment Arms	Endpoint	mDFS in Stage II-IIIA (Months)	2-Year Survival Without CNS Metastases (%)	5-Year OS (%)	Any G AEs (%)	G ≥ 3 AEs (%)	Dose Interruptions (%)	Dose Reductions (%)	Discontinuation (%)
3	IB-IIIA	Osimertinib vs. placebo	Primary: DFS among patients with stage II-IIIA Secondary: DFS in overall population, OS HRQoL, safety	NR vs. 19.6	98 vs. 85	88 vs. 78	98 vs. 89	20 vs. 13	24 vs. 11	9 vs. 1	11 vs. 3

HRQoL = health-related quality of life; NR = not reached; years = years; AEs = adverse events.

**Table 3 cancers-17-00668-t003:** Ongoing clinical trials on osimertinib in adjuvant setting.

Trial	Phase	Pathologic Stage	N. pts	Treatment and Schedule	Primary Endpoint	Secondary Endpoint	Status	Results Awaited
TARGET NCT05526755	2	II–IIIB	180 (150 cohort with common mutations, 30 cohort with uncommon mutations)	Osimertinib 80 mg/day for 5 years	DFS at 5 years in cohort with common mutations	DFS at 5 years in cohort with uncommon mutations cohort: OS at 3, 4, and 5 years in overall population; safety and tolerability; type of recurrence and CNS metastases	Active, not recruiting	2029
ADAURA2 NCT05120349	3	IA2–IA3	380	Osimertinib 80 mg/day vs. placebo for 3 years	DFS in the high-risk group	DFS in the overall population, OS, CNS DFS, and safety	Active, recruiting	2032
OSTAR NCT05686434	2	I with high-risk factors (solid and/or micropapillary component ≥ 10%, and/or airway spread).	65	Osimertinib 80 mg/day × 3 years	3-year DFS rate	DFS, 3-year OS rate, 5-year OS rate, OS, safety, HRQoL and symptoms	Active, recruiting	2029

DFS = disease-free survival; OS = overall survival; CNS = central nervous system; HRQoL = health-related quality of life.

**Table 4 cancers-17-00668-t004:** Summary of LAURA study in locally advanced NSCLC.

Phase	Stage	N. pts	Treatment Arms	Endpoint	mPFS (Months)	ORR (%)	DoR (Months)	Any G Adverse Events (%)	G ≥ 3 Adverse Events (%)
3	III	143	Osimertinib vs. placebo	Primary: PFS Secondary: OS, CNS PFS, ORR, DoR, HRQoL, safety	39.1 vs. 5.6	57 vs. 33	36.9 vs. 6.5	98 vs. 88	35 vs. 12

PFS = progression-free survival; CNS = central nervous system; ORR = objective response rate; DoR = duration of response; HRQoL = health-related quality of life.

**Table 5 cancers-17-00668-t005:** Ongoing clinical trials on osimertinib in locally advanced, neoadjuvant, and early-stage NSCLC.

Trial	Setting	Phase	Clinical Stage	N. pts	Treatment and Schedule	Primary Endpoint	Secondary Endpoint	Status	Results Awaited
NEOLA NCT06194448	Locally advanced	2	III, unresectable	70	Osimertinib × 8 weeks → CRT con × 6 weeks → osimertinib maintenance until to PD	PFS	ORR, DCR, OS, EFS, AEs	Recruiting	2028
PACIFIC4 NCT03833154	Localized	3	I–II	60	SBRT → osimertinib up to 3 years	4-year PFS	OS, safety	Active, recruiting	2028
GALAXY-02 NCT06383728	Neoadjuvant	2	II–IIIB EGFR-mutated squamous lung cancer	51	Single-arm osimertinib (80 mg/d, ≥9 weeks)	ORR, safety	MPR, R0 rate, PFS, OS, DCR, DoR	Active, recruiting	2030
NCT06018688	Neoadjuvant	2	IIA–IIIA NSCLC	44	Single-arm osim + aspirin × 2 months	MPR	DFS, ORR, DCR, pCR	Active not yet recruiting	2026
NCT05011487 (NOCE01)	Neoadjuvant	2	IIIA–B (N2) NSCLC	30	Single-arm osim 80 mg × 60 days + chemo cis + pem × 2 cycles	complete lymph node clearance rate	MPR, pCR, downstaging, DFS	Active, recruiting	2028

PFS = progression-free survival; ORR = overall response rate; DCR: disease control rate; OS = overall survival; EFS = event-free survival; AEs = adverse events.

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
