# Peer review of "Osimertinib in the Treatment of Epidermal Growth Factor Receptor-Mutant Early and Locally Advanced Stages of Non-Small-Cell Lung Cancer: Current Evidence and Future Perspectives"

_cancers, 2025, doi:10.3390/cancers17040668_

Round 1

Reviewer 1 Report

Comments and Suggestions for Authors

The treatment of EGFR-mutant non-small cell lung cancer (NSCLC) has been transformed by EGFR tyrosine kinase inhibitors (TKIs), with osimertinib now established as the standard of care in advanced and early-stage disease. Clinical trials like ADAURA and LAURA have demonstrated significant survival benefits, with osimertinib improving overall survival in adjuvant settings for resected stage II-IIIA tumors and delaying progression in locally advanced disease following chemoradiotherapy. The ongoing Neo-ADAURA study is exploring its potential in the neoadjuvant setting, where efficacy data remain uncertain. EGFR mutations, found in 13% of Caucasian and 50% of Asian patients, include common mutations like exon 19 deletions and exon 21 L858R, which are highly responsive to TKIs, and uncommon mutations with varying sensitivity. Resistance mechanisms, such as secondary T790M mutations and altered apoptotic pathways, remain a significant challenge. Efforts to address resistance include investigating biomarkers like circulating tumor DNA (ctDNA) to personalize therapy and combining treatments to enhance efficacy. Advances in molecular biology and new therapeutic strategies aim to overcome resistance and improve outcomes for patients with EGFR-mutant NSCLC. This is an interesting article and is imprtant in the field of lung cancer and targeted therapies. However, authors should address few minor comments.  

Minor comments:

1. The authors should provide a more detailed explanation of the mechanisms of resistance in both the introduction and discussion sections. This should include an overview of key resistance pathways and their clinical relevance, with a focus on how these mechanisms impact therapeutic efficacy.

2. The manuscript should address whether the regulation of AXL and low-fidelity polymerases is influenced by circulating tumor DNA (ctDNA) or if ctDNA itself regulates these polymerases.

3. The authors should discuss whether ctDNA plays a regulatory role in activating APOBEC enzymes. Addressing these recent advancements is critical for providing a comprehensive understanding of resistance mechanisms and their therapeutic implications in the manuscript.

All these above points have to be addressed and discussed in the manuscript.  

Author Response

Comment 1: The authors should provide a more detailed explanation of the mechanisms of resistance in both the introduction and discussion sections. This should include an overview of key resistance pathways and their clinical relevance, with a focus on how these mechanisms impact therapeutic efficacy.

Response 1: We thank the reviewer for the suggestion. We have added some sentences about the resistance mechanisms in the introduction section.

A further paragraph (number 5) was added before conclusion, and titled “resistance mechanisms to osimertinib”. We have extensively analysed and classified the various mechanisms of resistance to osimertinib, both primary and acquired, discussing the possible therapeutic strategies that can be used to overcome resistance, such as drug combinations acting against specific targets identified as resistance drivers. We emphasised that there are no literature data of resistance mechanisms in earlier stages of disease and that the data are derived from metastatic disease treated with osimertinib.

Comment 2: The manuscript should address whether the regulation of AXL and low-fidelity polymerases is influenced by circulating tumor DNA (ctDNA) or if ctDNA itself regulates these polymerases.

Response 2: As suggested by the reviewer, we performed a literature analysis, identifying some recent works on the role of AXL in development of resistance to osimertinib. We have discussed the available data, although these data always refer to advanced disease. Despite the extensive literature search we were not able to identify data concerning the interplay between ctDNA and AXL.

Comment 3: The authors should discuss whether ctDNA plays a regulatory role in activating APOBEC enzymes. Addressing these recent advancements is critical for providing a comprehensive understanding of resistance mechanisms and their therapeutic implications in the manuscript.

Response 3: Similarly, we reported the avalaible data on the role of APOBEC in the development of resistance to osimertinib; again, the data always refer to advanced disease. Despite the extensive literature search we were not able to identify data concerning the interplay between ctDNA and APOBEC.

Reviewer 2 Report

Comments and Suggestions for Authors

My comments:

  1. This review for Osimertinib in the treatment of EGFR mutant in early stage NSCLC is quite comprehensive and updated with detailed introduction on various clinical trials related to Osimertinib.
  2. There is only one sentence is confusing and needs to be revised. On page 10, line 399-401:

the pathological response has not never been validated as surrogate of OS and long-term survival data are lacking.

Author Response

Comment 1: This review for Osimertinib in the treatment of EGFR mutant in early stage NSCLC is quite comprehensive and updated with detailed introduction on various clinical trials related to Osimertinib.

Response 1: We thanks the reviewer for his comment.

Comment 2: There is only one sentence is confusing and needs to be revised. On page 10, line 399-401:

“the pathological response has not never been validated as surrogate of OS and long-term survival data are lacking.”

Response 2: According to suggestion of the reviewer, the sentence was modified: the pathological response has not yet been validated as surrogate of OS in lung cancer and mature data on long-term survival are lacking.

Reviewer 3 Report

Comments and Suggestions for Authors

1.  This review is essentially a summary of findings with osimertinib, a 3rd generation TKI against placebo.  Although authors claim in lines 58-67 that this 3rd generation TKI is superior to 1st and 2nd generation TKI's, they provide no comparative data in support.

2.  I recommend insertion of an additional Table comparing therapeutic efficacy of osimertinib with other studies conducted with the first and 2nd generation TKI's.  Otherwise, this review reads like an advertisement for  the newest TKI version for the treatment of EGFR mutant NSCLC.

3.  With this additional information, I believe that this review will be a useful addition to the scientific literature.

Author Response

Comment 1: This review is essentially a summary of findings with osimertinib, a 3rd generation TKI against placebo.  Although authors claim in lines 58-67 that this 3rd generation TKI is superior to 1st and 2nd generation TKI's, they provide no comparative data in support.

Response 1: We thank the reviewer for the suggestion. We specified that it was a statistically significant survival advantage. 

Comment 2: I recommend insertion of an additional Table comparing therapeutic efficacy of osimertinib with other studies conducted with the first and 2nd generation TKI's.  Otherwise, this review reads like an advertisement for  the newest TKI version for the treatment of EGFR mutant NSCLC.

Response 2: We thank the reviewer for the suggestion. An additional Table (Table 1) was added to compare the efficacy of osimertinib with other studies conducted with the first and second generation TKIs.

Comment 3: With this additional information, I believe that this review will be a useful addition to the scientific literature.

Response 3: We thank the reviewer for his considerations.

Reviewer 4 Report

Comments and Suggestions for Authors

his is a long and detailed review about osimertinib and NSCLC.It covered many papers about osimertinib and NSCLC.It will helpful to know about the all the trials of osimertinib in NSCLC.However,my suggestions listed below:

1,All parts should be simplified . For some unimportant results can only  be listed in tables .For some important results could be simplified such as ADAURA and LAURA. It is too long and complicated for the conclusion part .It should only emphasis the important results of in different parts.For example ,it is not suitable to talk about rechallenging in this part.

2,The topic and contents looks a little conflict.How to define early stage NSCLC? It should includes local advanced NSCLC or not ? I think it need a little modified in topic or contents.

Author Response

Comment 1: All parts should be simplified . For some unimportant results can only  be listed in tables .For some important results could be simplified such as ADAURA and LAURA. It is too long and complicated for the conclusion part .It should only emphasis the important results of in different parts.For example ,it is not suitable to talk about rechallenging in this part.

Response 1: We thank the reviewer for the suggestions.

The description of the ADAURA and LAURA studies has been simplified.

Two tables were added, one for ADAURA study and one for LAURA study, to better summarise data about both endopoints and results of the studies.

The conclusion was simplified. The sentences about rechallange of osimertinib in the conclusion were deleted; they were moved to section “open questions”.

Comment 2: The topic and contents looks a little conflict.How to define early stage NSCLC? It should includes local advanced NSCLC or not ? I think it need a little modified in topic or contents.

Response 2: We share the considerations of the reviewer: locally advanced stage represents a heterogeneous groups of tumors, with different therapeutic options and prognosis. Therefore, we agree that it is more appropriate to change the topic. Changes have been made in title, abstract, simple summary, introduction and conclusion.

Reviewer 5 Report

Comments and Suggestions for Authors

Dear Editor,

I was pleased to read the manuscript titled “Osimertinib in the treatment of EGFR mutant early stage NSCLC: current evidence and future perspectives” by Veccia et al. The review focused on the use of osimertinib in early-stage NSCLC. It has been reported that phase 3 clinical trials support the use of osimertinib both in the adjuvant setting and as the new standard of care in locally advanced disease. The results of the ADAURA study and the LAURA study were compared. We also evaluated studies that sought to identify potential biomarkers of response, such as circulating tumor DNA (ctDNA), to select patients who benefit most from osimertinib. Congratulations to the authors for their efforts. However, there have been many reviews on the subject and similar situations have been expressed. unfortunately, the review does not approach the subject from a different perspective.

Sincerely

Author Response

We acknowledge the reviewer's comments. The editor asked not to respond.

Round 2

Reviewer 3 Report

Comments and Suggestions for Authors

Thank you for addressing the concerns I raised in my initial review.

Reviewer 4 Report

Comments and Suggestions for Authors

Thank you for your revision.I agreed to be published in this form.

Reviewer 5 Report

Comments and Suggestions for Authors

Dear Editor,

Thanks for the corrections and additions. It can be seen that significant improvement has been achieved in the manuscript.

Sincerley